# Metabolic Basis of Circadian Dysfunction in Parkinson’s Disease

**DOI:** 10.3390/biology12101294

**Published:** 2023-09-28

**Authors:** Priya Rathor, Ratnasekhar Ch

**Affiliations:** 1Metabolomics Lab, CSIR—Central Institute of Medicinal & Aromatic Plants, Lucknow 226015, India; priyarathore9066@gmail.com; 2Academy of Council of Scientific and Industrial Research (ACSIR), Gaziabad 201002, India; 3School of Biological Sciences, Queen’s University Belfast, Belfast BT9 5DL, UK

**Keywords:** Parkinson’s disease, circadian rhythm, circadian dysfunction, metabolism

## Abstract

**Simple Summary:**

Parkinson’s disease (PD) is the second most common neurodegenerative disorder. Although genetic predisposition plays an important role, circadian dysfunction caused by chronic sleep disorders in response to the industrialization of modern society has increased the prevalence of PD. This review highlights the intricate interplay between circadian rhythm, cellular metabolism, and PD pathogenesis. Understanding the metabolic underpinnings of circadian dysfunction in PD could lead to identifying novel biomarkers for early diagnosis of PD and developing targeted therapeutic strategies to better manage and treat this complex neurodegenerative disorder.

**Abstract:**

Parkinson’s disease (PD) is one of the most common neurodegenerative disorders. The management of PD is a challenging aspect for general physicians and neurologists. It is characterized by the progressive loss of dopaminergic neurons. Impaired α-synuclein secretion and dopamine release may cause mitochondrial dysfunction and perturb energy metabolism, subsequently altering the activity and survival of dopaminergic neurons, thus perpetuating the neurodegenerative process in PD. While the etiology of PD remains multifactorial, emerging research indicates a crucial role of circadian dysfunction in its pathogenesis. Researchers have revealed that circadian dysfunction and sleep disorders are common among PD subjects and disruption of circadian rhythms can increase the risk of PD. Hence, understanding the findings of circadian biology from translational research in PD is important for reducing the risk of neurodegeneration and for improving the quality of life. In this review, we discuss the intricate relationship between circadian dysfunction in cellular metabolism and PD by summarizing the evidence from animal models and human studies. Understanding the metabolic basis of circadian dysfunction in PD may shed light on novel therapeutic approaches to restore circadian rhythm, preserve dopaminergic function, and ameliorate disease progression. Further investigation into the complex interplay between circadian rhythm and PD pathogenesis is essential for the development of targeted therapies and interventions to alleviate the burden of this debilitating neurodegenerative disorder.

## 1. Introduction

Circadian rhythm is an intrinsic time-keeping mechanism that exists in almost all organisms from bacteria to humans by synchronizing their biological processes with a nearly ~24 h day/night cycle. These biological oscillations help organisms adapt to daily environmental changes [1]. These self-sustaining biological oscillations can respond to environmental stimuli like temperature, nutrients, and light [2]. In mammals, many physiological processes are under circadian control, including sleep–wake cycles, feeding behavior, core body temperature, blood pressure, cognitive performance, glucose homeostasis, and various hormonal secretions. In mammals, circadian clocks are present in almost all cells and tissues, and the central control lies with the “master clock” in the suprachiasmatic nucleus (SCN) located within the brain’s hypothalamus. This master clock is the primary driver of circadian rhythms and is vital in synchronizing peripheral clocks found in various tissues like the liver, pancreas, skeletal muscle, intestine, and adipose. The peripheral clocks respond to internal signals influenced by nutrients and temperature, ensuring proper coordination of physiological processes throughout the body [3]. Maintaining a synchronized circadian rhythm is crucial for health [4]. The circadian rhythm regulates sleep–wake cycles, release of hormones, immunity, and body temperature. Given the significant impact of food-related cues on various biological processes, including circadian rhythm, it is not surprising that they play a prominent role in influencing these physiological processes (Figure 1) [3]. Additionally, the SCN, which serves as the central pacemaker for circadian rhythm, receives input from specialized light-sensitive intrinsically photosensitive retinal ganglion cells (ipRGCs) in the retina. These cells contain the photopigment and melanopsin and transmit light information to the SCN, helping to synchronize circadian rhythm with the external light–dark cycle [4]. Furthermore, circadian rhythms influence the expression of genes involved in neuronal activity and synaptic plasticity. Moreover, attention, memory, and reaction time are just a few cognitive abilities significantly influenced by circadian rhythms [5].

Disruption of circadian homeostasis can affect human health and increase the risk of neurodegenerative disorders, such as Alzheimer’s disease (AD), PD, and Huntington’s disease.

### 1.1. Transcription/Translation Feedback Loops (TTFL) Mechanisms in Mammals

Critical advances in genetics revealed that the molecular clock in mammals is generated by TTFL, that drives 24 h rhythms of biological oscillations. In mammals, the core TTFL clock is driven by transcriptional activators/proteins, CLOCK and BMAL1, and repressor proteins, PERIOD (PER) and CRYPTOCHOME (CRY). During the active phase, CLOCK and BMAL1 form heterodimeric complex viz., CLOCK: BMAL1, which activates transcription of the PER and CRY genes and other clock-controlled output genes. PER and CRY are converted into proteins in the cytoplasm. PER and CRY proteins heterodimerize in the cytoplasm and translocate to the nucleus to interact with CLOCK: BMAL1, inhibiting further activation of transcription towards the end of the rest phase. PER and CRY proteins are degraded with time through ubiquitin-dependent pathways, and the corresponding repression on CLOCK: BMAL1 is relieved. This cycle begins again with a 24 h periodicity (Figure 2) [6]. Over a 24 h period, these positive and negative regulations cause the clock gene expression to cycle.

However, the inhibitory role of the PER proteins is less clear [7]. The rhythmic expression of BMAL1 is further regulated by a secondary feedback loop comprising REV-ERBA α which inhibits while RORα proteins activate Bmal1 transcription (Figure 2) [8,9]. The CLOCK: BMAL1 complex plays a crucial role in initiating another potential TTFL component by regulating the transcription of the PAR bZIP genes. These PAR bZIP genes contain D-box elements in their promoters and encode various proteins, including D-box binding protein (DBP), hepatic leukaemia factor (HLF), thyrotroph embryonic factor (TEF), and nuclear factor interleukin 3 (NFIL3) [10].

In addition to these factors, chromatin remodeling plays a critical role in sustaining the cyclic transcriptional activity of the main TTFL system by rhythmic acetylation and deacetylation of the H3 and H4 histones, facilitated by clock-specific and ubiquitous histone-modifying proteins [11,12]. Recent research findings identify the existence of other metabolic oscillators of cellular metabolism in addition to the TTFL oscillator, and these cytosolic metabolic rhythms cross-talk with a transcriptional–translational canonical clock system to maintain circadian rhythm and affect the TTFL clock system and vice versa [13].

### 1.2. Circadian Control of Cellular Metabolism

Circadian rhythm regulates cellular metabolic pathways by serving daily metabolic demands required by an organism [14]. With remarkable technological advancements, metabolomics has evolved into a routine “omics” approach for the investigation of metabolism. Recently, several research groups have undertaken circadian metabolomics studies. To date, circadian metabolome have been extensively reported in various biological samples, including human serum, saliva [15], breath [16], urine [17], and tissues from various species. Importantly, these investigations show that a major portion of metabolome is under circadian control [18]. These metabolites exert their influence by modulating rhythmic gene expression through various mechanisms. They can activate proteins that directly interact with the core clock machinery or act on chromatin-modifying enzymes responsible for regulating rhythmic gene expression [19].

One well-studied example is the metabolite NAD^+^ known for its role as an activator of several NAD^+^-dependent sirtuin proteins. NAD^+^ is crucial in regulating the rhythmic circadian output at the gene expression level, specifically by influencing the CLOCK:BMAL1 transcriptional complex [19]. NAD^+^ also plays a role in orchestrating rhythmic activation of SIRT3 within the mitochondria [20]. This activation leads to oscillations in the acetylation and activity of downstream enzymes that play a crucial role in regulating mitochondrial oxidative function.

Some crucial metabolites which directly affect the circadian clock are better characterized. Notably, polyamines have recently been found to oscillate and provide feedback to the circadian clock system [21]. Polyamines, which have been observed to decrease with age, play significant and diverse roles within the cell. They can attach to DNA and proteins, controlling various functions from gene expression to ion channel operation. The circadian clock and feeding habits impact the rhythmicity in polyamine abundance [22]. Ornithine decarboxylase (ODC), the rate-limiting enzyme in polyamine production, experiences oscillations throughout the circadian cycle controlled by BMAL1: CLOCK trans-activation. It has been found that a longer circadian period is caused by polyamine depletion in cells and mice, principally because the PER2:CRY1 protein connection is compromised. Increasing polyamine concentration, however, has the reverse effect, and shortens the circadian period [21]. Additionally, considerable alterations were seen in urinary metabolites during sleep loss compared to regular sleep. Eight specific metabolites—taurine, formate, citrate, 3-indoxyl sulphate, carnitine, 3-hydroxyisobutyrate, TMAO, and acetate—showed increased levels, whereas eight additional metabolites—dimethylamine, 4-DTA, creatinine, ascorbate, 2-hydroxyisobutyrate, allantoin, 4-DEA, and 4-hydroxyphenylacetate—showed decreased levels in sleep deprived subjects [17]. In addition, the circadian metabolome and the rhythmic glycolysis and pentose phosphate pathway metabolites in human red blood cells highlight the essential role of cellular metabolism in their non-transcriptional circadian clocks. RBCs heavily depend on glycolysis pathway for ATP and pentose phosphate pathway for NADPH, which are crucial for redox balance and antioxidant protection. Throughout the circadian day time, the flux within the pentose phosphate pathway (PPP) achieves its peak while glycolysis operates at its lowest point. Conversely, glycolytic flux reaches its highest level at night, coinciding with the lowest PPP flux. The metabolic changes during these day and night cycles involve critical molecules such as Glucose-6-phosphate (G6P), Fructose-6-phosphate (F6P), Adenosine triphosphate (ATP), 2,3-Biphosphoglycerate (2,3-BPG), Glyceraldehyde-3-phosphate (G3P), Nicotinamide adenine dinucleotide phosphate reduced form (NADPH), and over-oxidized peroxiredoxin protein (PRDX-SO2/3) [23] (Figure 3).

## 2. Circadian Dysfunction and PD

PD is the second most prevalent chronic, progressive neurodegenerative disorder worldwide. Currently, for this disease there is no effective therapeutic intervention to delay or stop progression [24,25]. The main clinico-pathological feature is the loss of dopaminergic neurons (DANs) in the substantia nigra pars compacta (SNpc) and the presence of Lewy bodies [26]. Additionally, PD development has been linked to changes in the blood–brain barrier’s (BBB) integrity [27]. PD’s most common movement problems include stiffness, bradykinesia, and tremor while at rest. In addition, PD results in non-motor symptoms such as melancholy, olfactory dysfunction, digestive issues, cognitive decline, and sleep deficits [28]. Both motor and non-motor manifestations of PD can affect circadian rhythm [29]. As PD is predominantly associated with old age, understanding the influence of aging on the circadian clock becomes crucial in comprehending the effects of PD. Furthermore, old age is associated with alterations in sleep patterns, including sleep efficiency, decreased total sleep time, diminished REM sleep duration, increased sleep onset latency, periodic limb movement, and a higher arousal index [25]. Sleep loss can lower the quality of life, and according to earlier studies, people with PD have more severe sleep disruption [30].

### 2.1. PD and Metabolism

Alpha-synuclein in cerebrospinal fluid and dopamine metabolites have been the main subjects of biomarker research in PD. However, neither of these biomarkers are particularly useful for early identification, accurate prognosis, or tracking the course of the disease. Given the complex nature of PD, there is an urgent need for more precise diagnosis and individualized treatment to produce the greatest results. Metabolomics, which entails sophisticated metabolite profiling of bodily fluids like serum/plasma, CSF, or urine, has recently come to be recognized as a potent and advantageous approach for identifying novel biomarkers or metabolic fingerprints specific to PD at different stages of the disease [31]. Numerous investigations have consistently found alterations in the metabolism of alanine, branched-chain amino acids and fatty acids, and TCA cycle intermediates in PD, which suggest mitochondrial dysfunction. Additionally, investigations on metabolite profiling frequently reveal changes in the metabolism of purines (uric acid) and aromatic amino acids (phenylalanine, tyrosine, and tryptophan) in PD patients. Pyruvate emerges as a critical metabolite contributing to the distinct separation of PD samples from those of controls [32]. Gene expression analysis reveals significant alterations in PDHB and NPFF genes, leading to increased pyruvate concentration in blood plasma [32]. Overall, the combination of pyruvate and N8-acetyl spermidine and their respective associations with gene expression changes may provide important insights into the pathophysiology of PD [31]. The analysis of plasma samples revealed an elevation in pyroglutamate and 2-oxoisocaproate (ketoleucine) levels in PD patients, potentially indicating increased metabolic stress in these individuals. Conversely, in cerebrospinal fluid (CSF), a general decrease in metabolite levels, tryptophan, 3-hydroxyisovaleric acid, and creatinine was observed in PD cases compared to that of controls [33]. Furthermore, significant alterations in alanine, creatinine, dimethylamine, glucose, lactate, mannose, phenylalanine, 3-hydroxy-isobutyric acid, and 3-hydroxy-isovaleric acid were observed as a result of PD [34].

### 2.2. Circadian Metabolism Changes Related to PD

It is reported that circadian clock orchestrates cellular metabolism and the bidirectional relationship between clock and metabolism has been reviewed extensively elsewhere [14]. The convergence of PD with circadian metabolism offers a promising avenue for investigation, potentially bridging the gap in the current literature. Exploring the intricate connections between these two domains could lead to a more profound comprehension of PD’s underlying pathophysiology. The molecular mechanisms linking circadian disruption, altered metabolic function, and neurodegeneration are not fully understood yet and studies on circadian metabolic changes in PD models are very limited and few studies are discussed below.

#### 2.2.1. Circadian System and Dopamine

Dopamine plays a pivotal role in circadian regulation of dopamine metabolism. Dopaminergic neurotransmission is the core in PD related disorder, so it is important to evaluate the circadian variations caused by dopamine [35]. Daily rhythms in dopaminergic activity can be controlled by clock genes, which in turn, might also regulate activity of circadian clock itself. Studies reported that dopamine regulates the rhythm of clock protein, Per2 expression, by activation of DR22 receptors and striatal dopamine regulates CLOCK/BMAL1 heterodimer activity in a receptor-dependent manner. Circadian changes in dopamine could be directly linked to daily rhythms in transporters (DAT) and its synthesizing enzymes tyrosine hydroxylase (TH) as their activity exhibits temporal changes in cortical structures and basal ganglia. Indeed, timing of behavior could be directly correlated to the molecular mechanism of circadian rhythms in dopaminergic function [36,37]. Dopamine metabolism influences the underlying circadian fluctuations in symptoms associated with restless legs syndrome, which is a movement disorder of sleep associated with PD [36,37]. Alterations in circadian dopamine metabolism have been observed in a glutamate transporter 3 knockout mouse PD model [38]. Further, in a study conducted on mouse model of PD, deficits in locomotion and circadian dopamine metabolism were observed. Alteration in dopamine levels affects circadian rhythmicity at molecular and behavioral levels. Dopamine depletion in a chemical-induced PD model leads to altered BMAL1 levels and these levels are positively correlated with PD severity. It may be due to the dopamine capacity to regulate CLOCK/BMAL1 heterodimer activity [39]. These results clearly indicate that dopamine depletion may directly affect the central component of molecular clock and circadian disruption, that can accelerate PD progression.

#### 2.2.2. Circadian Dysfunction of Serotonin Metabolism

In addition to dopamine, serotonin (5-hydroxytryptamine, 5-HT) also plays a critical role in modulating other neurotransmitters, including glutamate and GABA, as well as feedback mechanisms in CNS, allowing it to perform basic physiological functions in the brain such as sleep, arousal, mood, and emotion [40]. In the brain, there is a dense serotonergic innervation of basal ganglia from dorsal raphe nuclei that send projections to the frontal cortex limbic system. In post-mortem studies of PD patients, depletion of serotonin levels in the frontal cortex, caudate, and hypothalamus were reported, suggesting the potential role of serotonin in PD. The SCN in the brain regulates the rhythmic production and discharge of serotonin via multi-synaptic efferent pathways and serotonin metabolism itself shows rhythmicity in SCN evident in rat model studies. Alterations in the daily rhythms of serotonin metabolism components including tryptophan, 5-hydroxytryptophan (5-HTP), 5-HT, N-acetyl serotonin, and melatonin were found in the SCN of a neurotoxin-induced PD rat model [41].

#### 2.2.3. Circadian Disruption of Energy Metabolism

Brain energy metabolism is the main determinant of neuronal viability. Glucose is metabolized by glycolysis, tricarboxylic acid cycle, and mitochondrial oxidative phosphorylation. These metabolic pathways play a crucial role in the pathogenesis of PD. Reduction in glucose metabolism and mitochondrial dysfunction have been observed in PD brains. The energy-producing ability of mitochondria is strongly associated with their morphology, abundance, and dynamics, which depend on the mitochondrial biogenesis, mitophagy, fission, and fusion, all shown to undergo circadian changes. Considering mitochondrial function rhythmicity and the crucial role of metabolic dysfunction in PD, very limited studies are available to understand the interplay between circadian clock disruption and energy metabolism. Circadian changes in energy metabolism were observed in fibroblasts (parkin mutation) derived from PD patients. Pacelli et al. [42] measured the oxygen consumption rate (OCR) and extracellular acidification rate (ECAR), which are direct indicators of mitochondrial respiratory and glycolytic activities. Parkin plays a critical role in regulating this energy metabolism. Parkin is a gene encoded by PARK2, shown to be involved in the mitophagy-mediated mitochondrial quality control. More than half of PD cases have been associated with mutations in the PARK2 gene coding for parkin, which is crucial for mitophagy-mediated mitochondrial quality control. In this study, dramatic damping of the time-dependent oscillatory profile of both respiratory and glycolytic activity were observed in PD fibroblasts (Figure 4). Furthermore, alterations in bioenergetic rhythmicity were observed in fibroblasts derived from PD patients (Figure 5). In supporting this, it has been reported that circadian rhythms of the bioenergetic metabolism precede clock gene expression in mouse embryonic stem cells [43]. Cellular ATP is an important energy source for many metabolic reactions. Adenosine mono phosphate activated protein kinase switches on catabolic pathways to generate ATP. It was reported that both ATP and AMPK show rhythmic expression. An in vitro study conducted by Hayashi et al. found loss of daily rhythms in ATP and AMPK in a chemical-induced PD model of SH-SY5Y cells treated with MPP+ for 48 h [44]. Activating AMPK and treating cells with ATP improves neuronal, behavioral deficit, and circadian dysfunction in PD.

#### 2.2.4. Circadian Disruption of Hormone Metabolism

It has been reported that the melatonin synthesis and the corticosteroid secretion are regulated by SCN in the brain and the output signals of these two hormones, considered as well-established markers of the central clock work to reflect endogenous rhythmicity. Melatonin is mainly produced in the pineal gland, during the dark phase, and regulates sleep–wake cycles. Two thirds of PD patients suffer from sleep disturbances and several studies have found altered circadian phase and reduced melatonin production with arrhythmic expression of serum melatonin levels [45]. Breen et al. reported that in early-stage PD, more than half of the patients have sleep problems with reduced melatonin production and arrhythmic expression. It was found that reduced melatonin production in PD patients is significantly related to degeneration of hypothalamic gray matter and the disease severity [46].

Cortisol is mainly produced by the hypothalamic pituitary adrenal (HPA) axis and the secretory rhythm of cortisol could be regarded as a sensitive marker for circadian function. Altered circadian rhythm of cortisol was found in PD patients. Hartmann et al. [47], measured the release of cortisol in plasma over 24 h and found that PD patients secrete significantly more cortisol, and the diurnal mean cortisol secretion rate is significantly higher but tends to be flatter than those in matched controls. Furthermore, hypercortisolism has also been found in patients with early PD. Breen et al. found elevated total serum cortisol levels with arrhythmic profiles [48]. The same phenomenon has also been found in the saliva of PD patients. These results provide information that crucial circadian markers, melatonin, and cortisol secretions were altered in PD subjects.

Collectively, these basic investigations provide support for showing alterations of the circadian metabolism in PD. Circadian clock has been shown to interact with a number of other metabolic pathways, which are critical for brain health and PD. Further studies will be needed to explore and understand the more detailed metabolic pathways of circadian disruption associated with PD.

### 2.3. Neural Basis of Circadian Dysfunction in PD

In mammals, the circadian system operates hierarchically, with the hypothalamic SCN as the master pacemaker. In addition to this, extra-SCN brain regions play a crucial role in regulating brain circadian rhythms [49]. Neural circuitry projecting signals from the SCN extend to various hypothalamic regions; these include the arcuate nucleus (ARC) [50,51], paraventricular nucleus (PVN), lateral hypothalamic area (LHA), dorsomedial hypothalamus (DMH) [52,53], as well as brain stem regions like the ventral tegmental area (VTA) and dorsomedial nucleus of the vagus (DMV) through intermediaries like the medial preoptic area (MPO) and PVN [54,55]. The PVN regulates the autonomic [54] while LHA the sleep–wake [56] cycles, The VTA and NAc are potential extra-SCN oscillators, and the neurons present in these regions exhibit circadian properties [57]. The circadian function of VTA and LHA might result from direct regulation by the circadian molecular clock, encompassing factors such as tyrosine hydrolase (TH) [58], dopamine transporter (DAT) [59], and the dopamine-degrading enzyme monoamine oxidase A (MAOA) [60]. PD involves the loss of dopamine-producing neurons in the substantia nigra, leading to motor issues and non-motor symptoms like sleep disturbances and neuropsychiatric problems [61]. Dopamine neurons from the VTA connect to the prefrontal cortex (PFC) and NAc, influencing motivation, emotions, and reward [62]. Additionally, the nigrostriatal pathway connects the SCN to the dorsal striatum and is crucial for motor functions [63].

Circadian rhythms are found in various brain regions beyond the SCN, including the hypothalamus [64], olfactory bulb [65], amygdala, cerebellum, cerebral cortex, and hippocampus [66]. The olfactory bulb has its own internal clock, more active at night, governing olfactory responsiveness [65]. The amygdala, including the basolateral and central nuclei, interacts with the memory-related hippocampus. These limbic regions have circadian gene activity, likely affecting cognitive performance rhythms [67]. Notably, PER2 gene expression differs in the amygdala and hippocampus. Experiments on animal models identified that the central amygdala peaks at night, while the basolateral amygdala and hippocampus peak in the morning. In PD regulation, complex neural circuits and various neurotransmitters are involved. Multiple brain regions influencing PD, such as the hippocampus, prefrontal cortex (PFC), ventral tegmental area (VTA), nucleus accumbens (NAc), amygdala, hypothalamus, and lateral habenula, communicate through dopaminergic (DAergic), nor-adrenergic, serotonergic, glutamatergic, and GABAergic pathways [68].

Understanding the relationship between the circadian clock, brain functions, and behavior control, including midbrain dopamine circuits, is important in normal and pathological conditions. Further, the SCN clock in the brain is influenced by the choroid plexus (CP). The entire brain is enveloped in cerebrospinal fluid (CSF), with a significant portion of this CSF production attributed to the CP. The CP plays a crucial role in regulating the circadian homeostasis of the brain by orchestrating the synchronization of “twist” circadian oscillators through gap-junctional connections [69]. By utilizing an in vitro tissue co-culture system and targeted deletion of the Bmal1 gene in vivo to suppress the CP’s circadian clock, it has been revealed that the CP’s circadian rhythm has a notable influence on the SCN clock. This influence, possibly facilitated by cerebrospinal fluid circulation, highlights the CP as an unexpected peripheral circadian clock, particularly intriguing given that non-neuronal tissues typically lack long-distance cell-to-cell signaling despite their significant size. Consequently, the non-neuronal network of circadian clocks within the CP emerges as an integral component of the brain’s circadian system, providing crucial feedback to the master SCN clock [69].

### 2.4. Circadian Dysfunction in PD Patients

The largest study on non-motor symptoms of PD found that 64% of patients had sleep issues [70]. People with PD frequently display a typical sleep macro-architecture characterized by increased day time sleep, decreased REM sleep length, decreased sleep efficiency, and increased sleep latency [48]. A slight delay in sleep onset time among PD patients (with a mean age at diagnosis of 68.0 years) compared to those of age-matched healthy controls was observed [28,48]. There are several circadian dysfunctions studies in PD mentioned in Table 1. Another important study on patients with PD identified a significant decrease in the amplitude of core body temperature rhythm compared to those of age-matched healthy controls mentioned in Table 1 [71]. Further studies identified that there is a decrease in the amplitude of melatonin oscillations in different body fluids, including plasma [30], serum [48], and saliva [28]. Furthermore, patients with PD may experience the loss of the typical circadian dip in blood pressure observed during the night [72]. This puts individuals with PD at a considerably higher risk of cardiovascular complications, including nocturnal hypertension [72].

In another study conducted on PD patients, it was observed that untreated patients with early PD (with a mean age of 66.3 years) had a doubled frequency of excessive daytime sleepiness compared to those of healthy controls at baseline. Furthermore, after five years of dopaminergic treatment, these patients exhibited a tripled frequency of excessive daytime sleepiness compared to those of the control group [29]. Recent research has shown that neurodegeneration affects the retina of PD patients, resulting in impaired retinal ganglion cells (RGCs) [73]. A sub-population of RGCs expressing melanopsin, a photopigment involved in circadian entrainment, is significantly reduced in PD. This reduction is correlated with poor sleep quality and thinning of the nerve fiber layer. The peripheral molecular clock, represented by the expression of clock genes such as BMAL1 in total leukocytes, is dampened in PD patients, indicating alterations in circadian rhythm. The relative levels of BMAL1 show a positive correlation with the severity of PD, suggesting that these molecular changes could serve as a potential basis for monitoring disease progression and evaluating the response to investigational drugs [74]. In most human studies, it was observed that there is loss of circadian oscillations in circulating melatonin levels and a reduced amplitude of melatonin rhythm in PD patients.

**Table 1 biology-12-01294-t001:** Studies for circadian dysfunction in human PD subjects.

S. No	Participants	Type of Circadian Markers	Measure of Circadian Markers	Results	Reference
1	169 age- and sex-matched controls and 153 drug-naive patients with Parkinson’s disease (mean age, 66 years)	Excessive daytime sleepiness (EDS)	Epworth sleepiness scale (ESS)	At baseline, 12% of PD patients and 5% of controls had EDS; after 5 years on PD treatment, 23% of PD patients and 8% of controls still had EDS.	[29]
2	20 PD patients and 15 controls of similar age (mean age 64 years)	Melatonin rhythm; EDS	24 h repeated blood collection to measure plasma melatonin; ESS	When compared to controls, patients with PD had a four-fold lower 24 h AUC for circulating melatonin levels and a reduced melatonin rhythm amplitude (*p* = 0.0001); there was no discernible difference in DLMO. EDS was seen in 27% of controls and 60% of PD patients (*p* = 0.01).	[30]
3	30 PD patients (mean age at diagnosis: 68 years) and 15 controls who were age and sex matched	Timing of sleep, peripheral clock gene expression, cortisol rhythm, melatonin rhythm, and EDS	14-day actigraphy, ESS, and 24 h repeated blood samples for serum melatonin and cortisol	In comparison to controls, patients with PD had lower levels of circulating melatonin (*p* = 0.005), higher levels of cortisol (*p* = 0.0001), and altered Bmal1 expression (*p* = 0.004). Patients with PD also showed more fragmented motor activity over the course of 24 h and later sleep start time.	[48]
4	28 age-matched controls and 29 patients with PD (mean age 642 years; 16 treated with medication, 13 not)	Timing of sleep, rhythm of melatonin, and phase angle of entrainment	salivary melatonin assay and 14-day actigraphy	The amount of melatonin secreted and the phase angle of entrainment were more than doubled (*p* = 0.001) in PD patients under dopaminergic therapy compared to controls, whereas there were no variations in sleep time or DLMO.	[28]
5	12 PD patients (mean age, 62 years) and 11 age-matched controls were studied.	Timing of sleep and profile of core body temperature	14-day actigraphy; 24 h ingestible capsule sensor used to record temperature profile	There was no change in sleep schedule that was statistically significant, although patients with PD had lower temperature mesors and lower nocturnal temperature amplitudes than controls.	[71]
6	111 PD patients, average age 67.8 years	Blood pressure	A 2 -h ambulatory blood pressure check	PD patients showed a high burden of nocturnal hypertension and 71% of them did not typically see a drop in blood pressure at night.	[72]
7	33 persons with Parkinson’s disease (age range: 52 to 72 years; mean age, SD)	Dim light melatonin	Melanopsin	Melanopsin-mediated post-illumination pupil response amplitudes were considerably decreased in PD (*p* = 0.0001) and linked with both nerve fiber layer thinning and poor sleep quality (r2 = 33 and 0.40, respectively, both *p* = 0.001). Higher subjective sleep ratings and earlier melatonin onset in people with Parkinson’s disease (PD) were both associated with significantly worse sleep quality (*p* = 0.05). Reaction of the outer retina to pupil lights the groups’ measurements, daily light exposure, and outer retinal thickness were comparable (*p* > 0.05).	[47]
8	17 patients	Clock gene	BMAL1 PER	During the dark period, BMAL1’s expression pattern differed, whereas PER1’s did not.	[74]

### 2.5. Circadian Dysfunction in the Animal Model of PD

In the context of PD, several animal models have played a pivotal role in advancing our understanding of the disease mechanisms [25]. Although sporadic PD instances with genetic links have been documented, they are rare. Risk factors from both the environment and genetics are thought to combine to cause late onset of idiopathic PD. Chronic stress has been proposed to be a significant environmental risk factor for the onset of neurodegeneration in PD.

#### 2.5.1. Environmental Toxin Models

##### 1-methyl-4-phenyl-1,2,3,6-tetrahydropyridine (MPTP) Model

MPTP is a neurotoxin known to induce dopaminergic neurodegeneration. Clock genes play a critical role in regulating neurological functions. One of the studies conducted on Bmal1 knockout mice resulted in the reduction of dopaminergic neurons along with microglial and astrocytic activation in the substantia nigra pars compacta and striatum. The loss of clock genes Bmal1 shows a 60% reduction in tyrosine hydroxylase (TH) protein levels [75]. Another study on MPTP-induced PD-like symptoms in a mouse model identified exacerbation in motor symptoms by causing a significant neuro-inflammatory response and dopaminergic neuronal degeneration [76]. Additionally, an increased inflammation in the brain was observed. These observations suggest that MPTP-induced PD mouse model exacerbates motor and cognitive deficits, dopaminergic neuronal loss, and neuro-inflammation. These results highlight the potential role of diet in influencing the development and progression of PD-related symptoms and neurodegeneration [76].

These findings are supported by one of the studies conducted on MPTP-induced PD mouse model, where loss of oscillations of the clock genes, Bmal1, Clock, Per1, Per2, Cry1, Dec1, and Rev-ErbA α was observed [44]. The study identified that the reduced number of dopaminergic neurons in the SCN would affect the circadian behavioural rhythms of mice. Further, loss of dopaminergic neurons and decreased tyrosine hydroxylase activity was observed. These findings established a link between age-related degeneration of the dopamine system, circadian behavioural rhythms, psychological deficiency, and the loss of dopaminergic neurons in the SCN [77,78].

##### Rotenone Model

Rotenone, a pesticide that inhibits mitochondrial complex I activity, mimics the features of early-phase PD and REM sleep behaviour disorder, dopamine-dependent behavioural deficits, hyposmia [26]. In addition to canonical pathological alterations, metabolic parameters like branched-chain amino acids, the tryptophan pathway, phenylalanine, and hyposmia were perturbed. Rotenone groups showed lower expression of the clock core genes, Bmal1, Clock, and neuronal PAS domain protein-2 (Npas2), Periods’ (Per1 and Per2) and Cryptochromes’ (Cry1 and Cry2). Additionally, there was a decrease in the expression of the molecular clock target genes nuclear receptor Rev-ErbA α (Nr1d1) and D-box-binding protein (Dbp) [79]. The expression levels of guanine nucleotide-binding protein subunit, lactate dehydrogenase, enolase, fructose bisphosphate, glycogen phosphorylase, and phosphoglycerate kinase vary in response to rotenone treatment [80]. These results are corroborated by a study by Mattam et al., who measured changes in the rhythmic dynamic equilibrium of interactions between various elements of serotonin metabolism and the molecular clock in the rotenone-induced PD male Wistar rat model. The findings showed a significant reduction in tryptophan, 5-hydroxytryptophan (5-HTP), serotonin (5-HT), N-acetyl serotonin (NAS), and melatonin (MEL) during a 24-h period [41].

##### 6-Hydroxydopamine (6-OHDA) Model

6-OHDA is another neurotoxin that causes the progression of PD-like symptoms in animal models. Gravotta et al., studied 6-OHDA-induced PD-like symptoms in the mouse model to investigate arrhythmic circadian components. Rats treated with 6-OHDA displayed pronounced sleep deficits, characterized by substantial disruption of circadian rhythms [81]. Furthermore, the authors found loss of BMAL1 rhythmicity in PD-like mouse models. These results demonstrate that Bmal1 regulates the inflammatory response by activating the NF-kB signaling pathways in macrophages, one of the most crucial transcription factors in the inflammatory response. LPS-stimulated glial cells are a major source of NF-kB activation [82]. A highly conserved class III deacetylase dependent on aldehyde dehydrogenase, SIRT1 is a member of the sirtuin protein family. The SIRT1 gene promoter’s gene polymorphism, potentially suppressing the gene’s transcription, may be a risk factor for PD. In support of the above findings, Boulamery et al. demonstrated disturbances in circadian rhythms of locomotion, temperature, and heart rate in a 6-OHDA-induced mouse model of PD-like symptoms [83]. In vitro and in vivo studies identified that treatment with 6-OHDA changes the expression patterns of clock genes and antioxidant molecules. Furthermore, altered expression in antioxidative genes and FABP4 in the PPAR signaling pathway was observed. Moreover, circadian rhythmicity of SIRT1 was lost [84,85]. SIRT1 is a significant clock component that involves the deacetylation of BMAL1 and PER2 proteins and binds to the CLOCK–BMAL1 complex to affect circadian rhythm and the expression of clock-controlled genes.

##### Manganese Model

Another neurotoxin is manganese, which inhibited rotarod performance, decreased the density of dopaminergic neurons in the substantia nigra, and activated microglia in the mouse brain [86]. In the brain and liver, exposure to Mn lowered the expression of clock-core genes in a dose-dependent way while increasing the expression of clock-targeted genes nuclear receptor Rev-ErbA α a (Nr1d1) and D-box-binding protein (Dbp). It appears that peripheral clock of the liver is more likely to be impacted by Mn-induced aberrant clock gene expression [86].

#### 2.5.2. Genetic Models

In addition to environmental models, the mutation in gene expression severely alters/affects dopamine release, causing dopamine neurodegeneration to attenuate PD. Researchers have developed genetic models like PARK7, PINK1, and SYN mutants for investigating the pathogenesis of PD. Studies conducted on *Drosophila* and mouse models of PD identified loss of circadian rhythmicity in clock genes, PER, TIM, and CLOCK [87,88,89]. Furthermore, circadian behavior and sleep loss were observed in these models. Excessive endoplasmic reticulum–mitochondrial contacts may influence sleep patterns and circadian dysregulation in PD models, but this can be alleviated in parkin and pink1 models by preventing excessive contact formation or enhancing phosphatidylserine levels [90]. In addition to these genetic models, researchers conducted studies on A53T and Mitopark genetic models of PD that showed reduced total sleep time and increased the sleep latency [91]. The summary of environmental toxin and genetic PD models were presented in Table 2.

## 3. Management of PD through Circadian Specialized Medicine or Circadian Physiological Intervention

Emerging evidence indicates that circadian oscillation disruptions are prevalent in PD, highlighting the potential for circadian regulation to be targeted in therapeutic interventions. Circadian therapy aims to restore the biorhythm by modifying external zeitgebers, primarily light intervention, physical activity, nutrient intervention, and social routines. By manipulating these external factors, it is possible to influence and potentially restore the circadian rhythm, offering a promising avenue for managing PD symptoms and improving overall well-being [97].

### 3.1. Light Intervention Can Manage PD

Chrono-therapeutic strategies targeting the circadian system by light exposure, melatonin supplementation, and timing of medication administration have shown promising results in improving sleep and overall symptom management in PD patients [98]. Paus et al. conducted a pilot study to evaluate the change in clinical symptoms in 36 patients receiving bright light therapy (BLT) [99]. In PD patients receiving dopaminergic treatment, bright light therapy enhanced sleep duration. It is possible to re-tick the biological clock with BLT intervention [73]. Additionally, utilization of BLT yielded positive outcomes in alleviating symptoms of bradykinesia and rigidity, consequently resulting in a successful reduction of dopamine replacement therapy in 50% of the participants involved in the study [39]. The systematic application of BLT was assessed in an open-label study involving 129 PD patients. The findings of this study demonstrated that BLT had a beneficial impact on the motor performance recovery of individuals with PD [89]. Furthermore, BLT is an excellent choice due to the benefits of non-invasion, low cost, and convenient usage.

### 3.2. Physical Exercise Can Manage PD

Exercise can affect the progression of PD in patients, in addition to serving as a non-photic time cue for the circadian clock. Exercise may improve the sleep–wake cycle, which acts independently of the circadian pacemaker [100]. Further evidence supporting the influence of physical exercise on circadian rhythms and its potential impact on disease progression has been investigated in animal models. For instance, studies have shown that voluntary wheel running, a form of physical exercise, can affect physiological circadian rhythms. Specifically, it has been observed to delay the phase of peripheral Per2 expression [101].

### 3.3. Nutrient Intervention Can Manage PD

The involvement of dopaminergic central nervous structures in regulating the circadian system suggests that the destruction of dopaminergic neurons by 6-OHDA can disrupt rhythm regulation [83]. However, a study indicated that continuous administration of L-Dopa, even for a period of seven days at a constant rate, improved and expedited the recovery of circadian rhythmicity [83]. In addition, it has been found that ATP improves circadian dysfunction and neuronal and behavioral deficits. Furthermore, SR8278, a REV-ERBA *α* antagonist, could modify the behavioral signs of the PD mouse model in a time-dependent way. In addition, the mechanism of action of SR8278 focused in particular on the crosstalk and genome-wide motif enrichment of REV-ERBA *α* and NURR1, another transcription factor implicated in the regulation of dopamine signaling [77]. Dietary addition of hesperidin, one of the antioxidant compounds in vegetables and fruits, controls the levels of the ROS, prevents DNA damage and neuro-inflammation, and restores rhythmicity [80]. Several studies [102,103] have indicated that dopaminergic treatment may induce a phase advance in the melatonin rhythm.

## 4. Challenges of Studying Circadian Metabolic Changes in PD

For the identification of an altered circadian metabolic system in PD models, it is important to consider the crucial steps involved in the process, starting from experimental design to identification of arrhythmic metabolites. Few important steps and their challenges are discussed below.

### 4.1. Selection of PD Model

Research indicates that circadian arrhythmicity is one of the key aspects which needs to be considered when investigating neurodegenerative disease, particularly PD. Circadian dysfunction and sleep disorders are central facets of PD care. To understand the altered circadian metabolic function, neurotoxin and genetic models are preferred.

Neurotoxin models are the most popular animal models to study circadian dysfunction in PD as these models can be made relatively easily. Although they mimic PD symptoms, neurotoxin models reflect the late, chronic, dopamine-depleted state. One of the main important limitations in animal PD models is the rapid and transient neurodegeneration. The major neurotoxins are MPTP, 6-OHDA, paraquat, rotenone [104]. Physiologically, 6-OHDA-induced models have been shown to exhibit circadian perturbations. Studies on animal models suggest that a 6-OHDA-induced PD-model reflects the disease state and sleep phenotype observed in human PD [25]. Although MPTP is highly lipophilic and has been regularly used for PD investigations, MPTP-treated mice established that the circadian dysfunction typically associated with PD is poorly reflected in this model. The circadian rhythm properties including free-running rhythms and entrainment remain intact in MPTP models. Furthermore, age-related circadian changes were not exacerbated in the MPTP neurotoxin model, more importantly at the molecular level. SCN clock gene profiles are rhythmic in MPTP-treated mice and both positive and negative TTFL components are unaltered, which questions the validity of this model. Furthermore, studies identified that circadian disruption in temperature, behavior and sleep dysfunction were observed in MPTP-treated mouse models. These findings suggest that the MPTP model could be used to investigate the sleep and circadian dysfunction, however, it is a less preferred model for studying molecular and metabolic investigations of circadian rhythms. Rotenone is a pesticide that induces PD-like symptoms by inhibiting mitochondrial respiratory complex 1. Studies on mice treated with rotenone showed significant decrease in the amplitude of core body temperature rhythms and progressive deterioration of NREM sleep [25]. Furthermore, alterations in daily rhythms of circadian clock gene expression of Per1, Per2, Cry1, and Bmal1 have been observed. These researchers found both behavioral and molecular changes in a rotenone-induced PD model. This neurotoxin would help to investigate circadian metabolic changes in cases of PD.

Different mutations in specific genes that cause mitochondrial dysregulation and energy disturbances are thought to cause PD. Several PD-linked genes, α-synuclein, Parkin, PINK1, and DJ-1, have been used for studying PD [25]. Mutation in α-synuclein in a mouse model found selective deficits in wheel running rhythms and these changes were progressive and age dependent [88]. It was found that peak and trough expressions of clock gene PER2 in the SCN did not differ and no evidence was found that circadian system was compromised in this model. Few studies found that circadian rhythm generation is not directly affected by the over-expression of α-synuclein. Additional research into this model is required to confirm the altered circadian function. The circadian phenotypes of Drosophila models with homologue knockouts of PINK1, PRKN, and DJ1 have been characterized. PINK1 and PRKN flies show disrupted circadian function, while in DJ-1 mutant flies no abnormalities were observed in the circadian rhythmicity [105].

### 4.2. Experimental Harvesting

Time of the day is crucial for experimental sampling in circadian metabolism monitoring experiments. Experimental evidence from animal models and human studies suggests that cellular clocks know the time of the day and the physiological function is controlled by the circadian clock. Sampling time is crucial in PD models as reports suggest excessive daytime sleepiness in PD patients compared with those of matched controls [106]. More importantly, the mammalian cellular metabolism has a specific molecular timetable around the day and the time at which metabolic parameters are measured can affect experimental results [17]. A majority of the circadian experiments with PD models are conducted in day light hours. This time frame corresponds to human’s diurnal behavior. However, rats and mice are nocturnal in nature and have a rest phase in the morning. Comparison of these daylight animal studies with human studies may provide wrong results, especially in the case of human PD subjects, who have different sleep and behavioral patterns during their daily cycles [106]. In addition to time of harvesting, sampling resolution plays a critical role in identifying circadian-regulated and perturbed metabolites in PD models. Higher temporal resolution is required in detecting metabolite rhythms. Furthermore, at least two cycles (i.e., 48 h) having repeated peaks and troughs need to be considered for a true circadian metabolism experiment [107]. More specifically, in case of PD, the circadian metabolism varies with time, for circadian metabolomics experiments, a tight resolution of one hour with two cycles is considered as a gold standard.

### 4.3. Analysis of Altered Circadian Metabolome in PD

Transcriptomics and genomics studies have revealed information on mechanisms regulating gene expression in PD models. Global metabolomics experiments with optimum temporal and spatial resolution can help us to understand altered circadian metabolism in PD models by revealing whole pathways of related metabolites. Although sample preparation procedures and selection of analytical platform are widely reviewed for metabolomics studies, these studies are limited to general metabolomics but do not deal with global circadian metabolomics to understand the altered circadian metabolism in PD models [108]. There are a few important things which need to be considered for circadian metabolomics studies, specifically in studying PD metabolism. Studying altered circadian metabolome experiments of PD models requires a large number of samples with high temporal resolution and a greater number of replicates and different chronological conditions. For example, measuring circadian metabolome in the brain of PD models requires accurate sampling of brain tissues and in some cases their components, such as SCN slices over multiple time points, as it is a challenging task due to tissue sensitivity and limited sample size.

Recent metabolomics studies suggest that many metabolites are specific to the organelle in which they reside [109]. Spatial metabolomics in PD models may provide information related to different physiological roles of metabolites depending on their intracellular localization. Most of the circadian metabolomics studies reported to date have not taken into consideration information related to subcellular compartmentalization of metabolites, for example, mitochondria, cytosol pools. Studying alteration in intracellular metabolite distribution in subcellular compartments can provide a clear picture of a perturbed circadian metabolism [109]. However, the time-consuming steps involved in organelle purification prior to metabolomics steps may likely introduce confounders. Instead, mass spectrometry techniques such as matrix-assisted laser desorption ionization coupled to imaging mass spectrometry is proving to be a promising technology for mapping the distribution of metabolome within tissue sections in situ.

Another important factor is circadian metabolome variability between same individuals during sampling procedures that creates technical variation and bias, which subsequently affects biological results. For example, it is nearly impossible to collect multiple SCN slices from different mice brains at the same time. Similarly, it may not be possible to collect multiple urine or blood samples at the same time over multiple circadian cycles. These technical variations can be eliminated by pooling samples together at the same time points and averaging out variation between individuals [107]. Measuring an independent circadian quality control must be considered when studying altered PD metabolism. Positive control can be done using clock genes or proteins or metabolites [107]. Another important factor is the volume of biological sample considered for measuring the altered circadian metabolome in PD models or subjects. It is possible to measure the blood samples from the same subject over multiple time points to remove variation. However, in the case of mice samples, the volume of blood collected is less than 0.9 mL, so uniform sampling should be performed from multiple animals and pooled blood samples from these animals can be used for circadian time series metabolomics experiments [107].

Analysis of altered PD metabolome is exceptionally challenging due to the chemical complexity and wide variety of metabolite profiles in master and peripheral clock tissues. The most common metabolomics tools are nuclear magnetic resonance spectroscopy (NMR) and mass spectrometry (MS). MS-based methods have the advantage of higher sensitivity and have a larger metabolome coverage with good peak resolution when connected with various chromatographic techniques. However, NMR has high reproducibility and is more quantitative. Mass spectrometry combined with a chromatographic system, such as liquid chromatography or gas chromatography, is the most widely used technology for studying the circadian metabolome [13]. One must consider important parameters like column chemistry, analytical procedures for analyzing altered PD metabolome. HILIC column chemistry with a gradient phase would be more suitable as many metabolites in the brain and peripheral tissues are polar in nature [110]. A recent promising study performed on mouse models has identified the system-wide coordination of metabolic communication in various tissues, where authors employed LC-MS based untargeted metabolomics to investigate circadian metabolism [111]. A similar approach would be helpful in studying altered circadian metabolism in PD models.

### 4.4. Identification of Altered Circadian Rhythmicity of Metabolites

Successful identification of altered circadian metabolites in PD models requires statistical tests that can identify phase, period, and amplitude of oscillation from large metabolomics datasets. Most routinely used algorithms to identify rhythmicity of metabolites from large metabolomics datasets are Wemen et al. [112]. (i) JTK_CYCLE: it characterizes metabolites as rhythmic or non-rhythmic using a non-parametric method based on a combination of the Jonckheere–Terpstra test [113]. (ii) ARSER: a harmonic regression based on autoregressive spectral estimation and combines time-domain, frequency-domain analysis to identify periodic metabolites. (iii) RAIN: a non-parametric method based on the time series ANOVA and cosinor analysis. (iv) Empirical JTK_Cycle: it was built based on the strengths of JTK_Cycle, however, eJTK_Cycle improves the performance of JTK_Cycle by calculating the null distribution such that it accounts for multiple hypothesis testing and by including non-sinusoidal reference wave forms. (v) MetaCycle: it is an alternative algorithm to detect rhythmicity in periodic data and incorporates ARSER, JTK_Cycle, and analyses single time series data and integrates their results into a common output file. (vi) Lomb–Scargle (LS): detects oscillations by comparing the data to sinusoidal curves of varying periods and phases [114]. These statistical tests can be used to identify the altered circadian metabolome in PD models.

## 5. Conclusions and Future Perspectives

Disruptions in the expression and functioning of crucial circadian clock genes and alterations in the dopaminergic system contribute to circadian disturbances in PD. These disturbances can cause nocturnal motor symptoms and disruptions in circadian patterns of hormone secretion. Studies on human PD subjects identified the loss of circadian oscillations of clock genes, Bmal1, PER and serum melatonin, cortisol, and melanopsin levels. Further, studies on animal models of PD revealed the loss of rhythmicity in genes, Bmal1, Cry1, and REV-ERB A α and dopamine, tryptophan, phenylalanine, and mitochondrial metabolites. More importantly, altered circadian function of dopamine metabolism, energy metabolism, and hormone metabolism were observed in PD models. It is crucial to conduct more extensive studies involving additional circadian markers to better understand the dopaminergic system and the role of metabolism in circadian rhythms in relation to neurodegeneration. Furthermore, there is a lack of a circadian blueprint of metabolic pathways either in human or animal models of PD to understand the circadian loss of metabolic oscillations. Understanding the underlying metabolic mechanisms of circadian dysfunction in PD may provide new avenues for therapeutic interventions to improve circadian regulation and overall disease management. Further research is needed to unravel the complex interplay between circadian metabolism and PD pathology, which may help to develop specific drugs for this neurological disorder.

## Figures and Tables

**Figure 1 biology-12-01294-f001:**
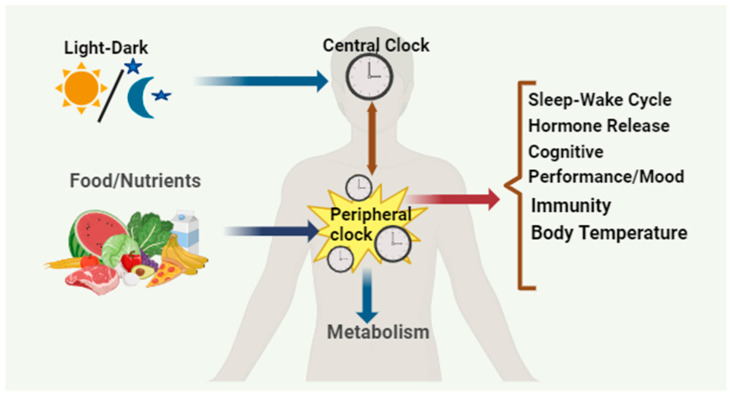
The circadian system exhibits a hierarchical organization. The master clock located in the SCN of the brain gets synchronized with the solar cycles and, in turn, synchronizes the behavioral and physiological rhythms throughout the body. This coordination extends to the circadian clocks situated in peripheral tissues. Additionally, feeding–fasting cycles play a role in entraining the clocks in metabolic tissues independently of the SCN’s neural activity rhythms.

**Figure 2 biology-12-01294-f002:**
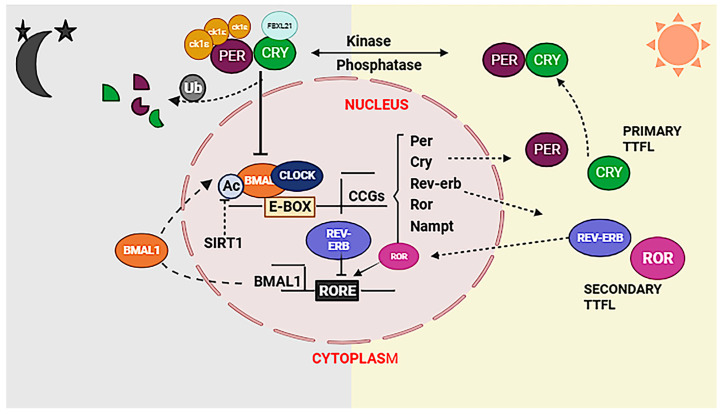
The proper functioning of the circadian system in a mammalian cell relies on the essential cross talk between transcription and metabolic cycles. The TTFL model involves core clock genes, BMAL1 and CLOCK, heterodimerizing to activate transcription factors PERIOD (PER) and CRYPTOCHROM (CRY) in active phase. Once in the cytosol, phosphorylation of PER and CRY leads to the formation of a complex that inhibits BMAL1/CLOCK-driven transcription, establishing a negative feedback loop in rest phase. This intricate interplay between transcriptional and translational cycles potentially enables coupling with metabolic cycles, creating a tightly regulated and coordinated connection between circadian rhythms and cellular metabolism.

**Figure 3 biology-12-01294-f003:**
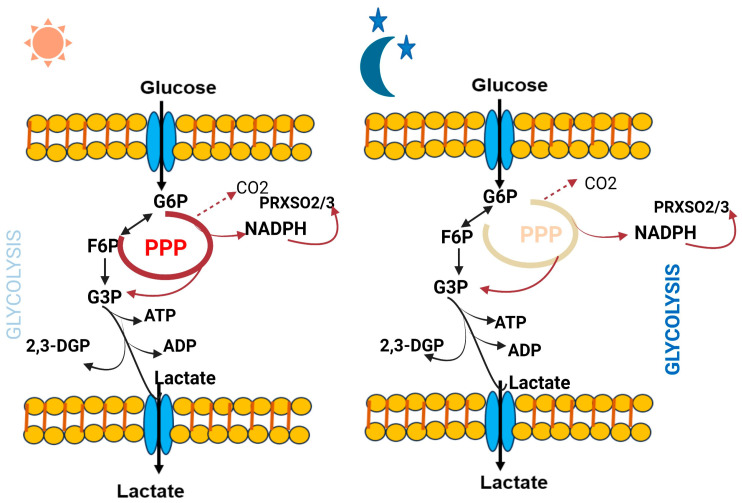
Dynamic changes in red blood cell glucose metabolism throughout the circadian day and night reveals intriguing insights into the rhythmic regulation of this essential process.

**Figure 4 biology-12-01294-f004:**
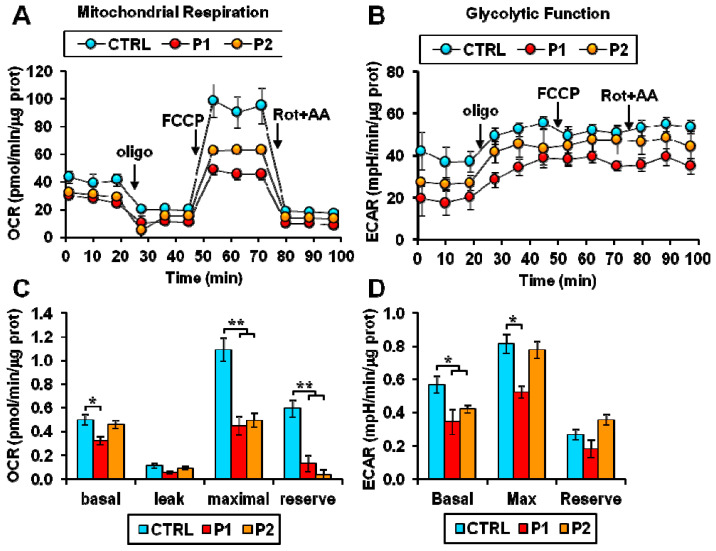
Altered energy metabolism in patient fibroblasts. (**A**,**B**) Representative Seahorse XF trace of oxygen consumption rate (OCR) (**A**) and extracellular acidification rate (ECAR) (**B**) performed in normal (CTRL) and Parkinson’s disease (PD) patient (P1, P2) fibroblasts. (**C**) Basal: resting OCR; leak: OCR measured in the presence of oligomycin; maximal: OCR measured in the presence of FCCP; reserve: difference between maximal and basal respiration. (**D**) Basal: resting ECAR; Max: ECAR measured in the presence of oligomycin and FCCP; Reserve: difference between maximal and basal glycolysis. * *p* < 0.01, ** *p* <0.005. Figure was adapted with permission of Pacelli et al. [41].

**Figure 5 biology-12-01294-f005:**
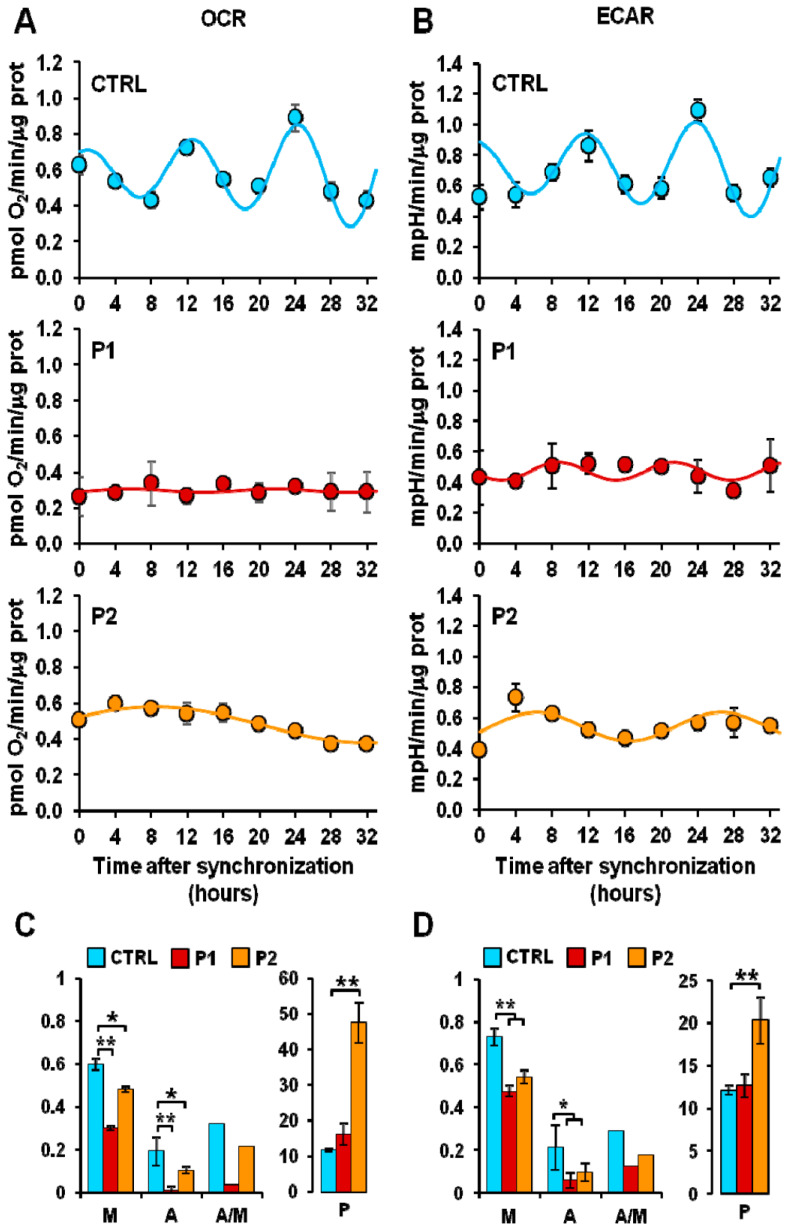
Altered mitochondrial respiratory and glycolytic activity in fibroblasts. (**A**) Resting oxygen consumption rate (OCR) and (**B**) basal extracellular acidification rate (ECAR) in serum shock-synchronized normal fibroblasts (CTRL) and PD patient-derived fibroblasts (P1, P2)—the time-course data points were best fitted with the COSINOR function. (**C**,**D**) The histograms display the mesor (M-x axis), the amplitude (A-x axis), and the A/M ratio (x-axis) for respiration (**C**) and glycolysis (**D**) of the best-fitting COSINOR. * *p* < 0.01, ** *p* < 0.005. Figure was adapted with permission of Pacelli et al. [41].

**Table 2 biology-12-01294-t002:** Circadian dysfunction studies in environmental and genetic PD model systems.

S. No	PD Model	Animal	Experimental Method	Circadian Rhythms Changes	Ref.
1.	MPTP model	Mice	qRT-PCR, WB, IHC	Mice lacking the BMAL1 gene treated with MPTP show a 60% reduction in tyrosine hydroxylase (TH) protein levels.	[75]
2.	MPTP Model	Mice	IHC, Behavior analysis	Circadian disruption causes greater loss of TH cell content and intense neuroinflammation.	[76]
3.	MPTP Model	Mice	Bioluminescence, RT-qPCR	Activation of AMPK results in circadian disruption, according to Bmal1, Cry1, and Rev-ErbA α a.	[44]
4.	MPTP Model	Mice	Behavior analysis and IHC	Lengthened free-running period.	[77]
5.	MPTP Model	Non-human Primates	IHC, Proteomics	Alteration in circadian rhythms (not significantly).	[78]
6.	Rotenone Model	Rat	Chronic Sleep Restriction	Affected a number of behavioral (reversal of locomotor activity impairment; cognitive impairment; delay of rest-activity rhythm) and metabolic (branched-chain amino acids, tryptophan pathway, phenylalanine, and lipoproteins, pointing to mitochondrial impairment) measures.	[26]
7.	Rotenone Model	Rats	Substantia nigra RT-qPCR, WB	Bmal1, Clock, NPAS2, Per 1 and 2, Rev-ErbA α a, and DBP. In RIPD rats, chronic low-grade neuroinflammation worsens circadian disruption.	[79]
8.	Rotenone Model	Rat	Behavior analysis	Reduced rhythm amplitudes and increased fragmentation in rhythm.	[92]
9.	Rotenone Model	Rat	qRT-PCR, WB, IHC	Lowered rhythm amplitudes, altered expression of clock genes, and increased rhythm fragmentation.	[41]
10.	6-OHDA Model	Rat	IHC, Constant dark	Activity decline and circadian activity rhythm interruption.	[81]
11.	6-OHDA model	Rat	Dopamine and levodopa measurement	Loss of circadian rhythmicity or changes.	[83]
12.	6-OHDA Model	Rat and neuroblastoma cells	Striatum for RT-qPCR; WB	Through SIRT1-dependent BMAL1 pathways, dysfunction of the circadian clock contributes to an aberrant antioxidant response in PD.	[93]
13.	6-OHDA Model	Rat	Striatum, SCN Plasma RT-qPCR, ELISA, HPLC	Bmal1 decrease, peak of Per2 delayed, cortisol secretion increased, and melatonin level decreased after levodopa treatment.	[85]
14.	6-OHDA Model	Rat	Immunostaining RT-PCR, HPLC	The frequency of dopaminergic activation of D2 DA receptors determines the rhythm of PER2 expression in the dorsal striatum.	[94]
15.	Mn2+	Rat	Hypothalamus, RT-qPCR, IHC	A few examples are an increase in Nr1d1 and DBP and a decrease in Bmal1, Clock, NPAS2, Cry1, Per1, and Per2.	[86]
16.	A30P	Drosophila	Behavior	Total amount of sleep is significantly reduced.	[95]
17.	PARK and PINK1 mutant	Drosophila melanogaster	RT-qPCR, WB IHC, LIPID	Greater sleep fragmentation and lower circadian power. Phosphatidylserine from the endoplasmic reticulum (ER) and disrupts the production of neuropeptide-containing vesicles.	[90]
18.	Mutant -SYN (A53T)	Mouse	EEG	Decreased total sleep time and NREM sleep.	[87]
19.	ASO Transgenic	Mice	IHC	The SCN of ASO mice do not exhibit changed Per2 expression, and PD is characterized by diminished circadian output.	[88]
20.	Mul 1A6 and Park1 mutants	Drosophila	RT-qPCR, IHC, WB	Per, Tim, and Clock’s typical circadian rhythmic expression during the day is interfered with by Mul 1 and Park mutations decreased ATG5.	[96]
21.	Mitopark	mouse		Increased sleep latency.	[91]

## Data Availability

The data presented in this study are available in this review article.

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
