# Peer review of "Metabolic Basis of Circadian Dysfunction in Parkinson’s Disease"

_biology, 2023, doi:10.3390/biology12101294_

Round 1

Reviewer 1 Report

PD disturbs sleep and circadian rhythms. This review article aims to dissect out the complex interplay between the circadian clock and PD. Overall the manuscript is well written. However, the authors should elaborate more on the neural basis of circadian disruption in PD. In particular, the midbrain  dopamine  reward system and several limbic structures appear to have endogenous clocks, or at least robust clock gene expression. They, in addition to the suprachiasmatic nucleus, are most probably involved in the interactions between PD and circadian rhythm disruptions. For example, dopamine neurons could have a critical role in the expression of sleep/wake and other rhythms at the systems level. The authors need to clearly point out the importance of those extra-SCN structures in the main text.

Author Response

Reviewer 1

PD disturbs sleep and circadian rhythms. This review article aims to dissect out the complex interplay between the circadian clock and PD. Overall the manuscript is well written. However, the authors should elaborate more on the neural basis of circadian disruption in PD. In particular, the midbrain  dopamine  reward system and several limbic structures appear to have endogenous clocks, or at least robust clock gene expression. They, in addition to the suprachiasmatic nucleus, are most probably involved in the interactions between PD and circadian rhythm disruptions. For example, dopamine neurons could have a critical role in the expression of sleep/wake and other rhythms at the systems level. The authors need to clearly point out the importance of those extra-SCN structures in the main text.

Response: Thank you for your positive response and your suggestions to improve the quality of the manuscript. All the suggestions have been implemented in the manuscript specifically the importance of extra-SCN structures is discussed under new section 2.2 Neural basis of circadian disruption in PD in the revised manuscript as per the suggestion

Reviewer 2 Report

The paper “Metabolic basis of circadian dysfunction in Parkinson’s disease” by Rathor and Ch aims to review the possible metabolic basis of circadian dysfunction that may lead to Parkinson’s disease (PD).

The paper is divided into 4 main sections: a) a general introduction to circadian rhythms; b) circadian rhythms and PD, both in patients and animal models; c) circadian treatments for PD. Although it is clear that an adequate background of circadian rhythms is needed to understand the paper, having the first 9 pages of the paper devoted to a general review of the molecular basis of the circadian system might be too much.

I am somewhat puzzled by the focus of the paper. To start with, a quick search in PubMed leads to more than 100 reviews in the last 5 years with the keywords “circadian” and “Parkinson”. The title of the present manuscript promises to deliver a new perspective on the subject, focalizing on circadian rhythms in metabolism and its putative link to the pathophysiology pf PD; however, this is not really achieved by the review. Instead, after a general introduction of the molecular and metabolic basis of the circadian system, the authors consider general features of PD in the clinic and in the lab. 

The two main tables might be an interesting addition to the literature. While Table 1 summarizes 8 clinical studies of PD, emphasizing sleep/circadian effects, Table 2 mentions 21 studies in animal models, describing changes in clock genes and circadian rhythms.

In general, the review lacks focus and could be greatly improved if the authors stick to their aim of relating PD with circadian metabolism. In this sense, it would need a complete rewriting in order to fill a relative gap in the existing literature.

Finally, the general language and grammar of the paper need to be carefully addressed so that the manuscript can be read much more clearly.

Extensive revisions are needed.

Author Response

The paper “Metabolic basis of circadian dysfunction in Parkinson’s disease” by Rathor and Ch aims to review the possible metabolic basis of circadian dysfunction that may lead to Parkinson’s disease (PD).

The paper is divided into 4 main sections: a) a general introduction to circadian rhythms; b) circadian rhythms and PD, both in patients and animal models; c) circadian treatments for PD. Although it is clear that an adequate background of circadian rhythms is needed to understand the paper, having the first 9 pages of the paper devoted to a general review of the molecular basis of the circadian system might be too much.

I am somewhat puzzled by the focus of the paper. To start with, a quick search in PubMed leads to more than 100 reviews in the last 5 years with the keywords “circadian” and “Parkinson”. The title of the present manuscript promises to deliver a new perspective on the subject, focalizing on circadian rhythms in metabolism and its putative link to the pathophysiology pf PD; however, this is not really achieved by the review. Instead, after a general introduction of the molecular and metabolic basis of the circadian system, the authors consider general features of PD in the clinic and in the lab. 

The two main tables might be an interesting addition to the literature. While Table 1 summarizes 8 clinical studies of PD, emphasizing sleep/circadian effects, Table 2 mentions 21 studies in animal models, describing changes in clock genes and circadian rhythms.

In general, the review lacks focus and could be greatly improved if the authors stick to their aim of relating PD with circadian metabolism. In this sense, it would need a complete rewriting in order to fill a relative gap in the existing literature.

Finally, the general language and grammar of the paper need to be carefully addressed so that the manuscript can be read much more clearly.

Response: Thank you for your positive response and your suggestions to improve the quality of the manuscript. All the suggestions have been implemented in the manuscript as per the suggestion.

As per the suggestion, we have reduced review of the molecular basis of the circadian system to five pages dealing core TTFL clock and circadian metabolism.

Further, we have tried to improve the manuscript specifically by focusing on PD and cellular metabolism in section 2 as suggested. Moreover, the general language and grammar of the manuscript was improved as per the suggestion.

Reviewer 3 Report

This review paper focused on the intricate interplay between circadian rhythm, cellular metabolism, and Parkinson’s disease (PD) pathogenesis. Simple summary indicates that understanding the metabolic underpinnings of circadian dysfunction in PD could lead to identification of novel biomarkers, enhanced early diganosis, and the development of targeted therapeutic strategies to better manage and treat this neurodegenerative disorder. The significance and complexity of circadian rhythm dysfunction in PD have been reviewed in great detail. These data are valuable and useful for scientific readers. The reviewer just has some comments. 

1)      Myung J, Takumi T, et al. reported that the choroid plexus is an important circadian clock component (Nat Commun 2018, 9: 1062). The reviewer should cite this paper and discuss on roles of the choroid plexus in circadian rhythm formation.

2)      Some abbreviations should be explained in the text, where they first appeared. Did SCN, EDS, and ESS explained in the text, where they first appeared?

3)      There are some typing mistakes (lines 386, 503, 644, 645).

Author Response

Reviewer 3

This review paper focused on the intricate interplay between circadian rhythm, cellular metabolism, and Parkinson’s disease (PD) pathogenesis. Simple summary indicates that understanding the metabolic underpinnings of circadian dysfunction in PD could lead to identification of novel biomarkers, enhanced early diganosis, and the development of targeted therapeutic strategies to better manage and treat this neurodegenerative disorder. The significance and complexity of circadian rhythm dysfunction in PD have been reviewed in great detail. These data are valuable and useful for scientific readers. The reviewer just has some comments. 

Response: Thank you for your positive response and your suggestions to improve the quality of the manuscript. All the suggestions have been implemented in the manuscript as per the suggestion. A

1)      Myung J, Takumi T, et al. reported that the choroid plexus is an important circadian clock component (Nat Commun 2018, 9: 1062). The reviewer should cite this paper and discuss on roles of the choroid plexus in circadian rhythm formation.

As suggested roles of choroid plexus in circadian rhythm formation has been discussed in under section 1.1 and the suggested paper was cited (reference 17) in the revised manuscript.

2)      Some abbreviations should be explained in the text, where they first appeared. Did SCN, EDS, and ESS explained in the text, where they first appeared?

As per the suggestion, abbreviations are explained in the text where they first appeared in the revised manuscript. 

3)      There are some typing mistakes (lines 386, 503, 644, 645).

Many thanks for the suggestion. We have corrected them in the revised manuscript as per the suggestion.

Round 2

Reviewer 2 Report

The authors have made an effort in order to make the article more focused on the theme they refer to in the title. However, I am afraid I am not still convinced about the relevance and originality of the review, considering that there are many papers on the subject. 

Additionally, the authors have marked changes in the new version of the manuscript, but have not provided a list of the precise changes and the rationale for them. 

It would certainly help if the authors identified the challenges of studying circadian changes related to PD and, in particular, the metabolic aspects of such relationship. A critical approach, more than a descriptive one, is needed here. 

While the authors have made some changes, the general legibility of the manuscript could be greatly improved by both a revision of the text and maybe running some kind of readability program, in order to make sure that the reading experience is smooth and (why not) enjoyable.

Author Response

Response to Reviewer 2:

Ref: Biology- 2559259

Title: Metabolic basis of circadian dysfunction in Parkinson’s disease

Dear Reviewer,

Many thanks for your valuable suggestions. We have tried best to improve the manuscript by implementing all the suggestions in the revised manuscript and response to each comment are mentioned below.

Thank you

Responses to comments:

  1. The authors have made an effort in order to make the article more focused on the theme they refer to in the title.

Response: We thank reviewer for the positive comment that we have made efforts in the first revision to present the article more focused on the theme titled” Metabolic basis of circadian dysfunction in Parkinson’s disease”.

  1. However, I am afraid I am not still convinced about the relevance and originality of the review, considering that there are many papers on the subject. 

Response: We have revised the manuscript according to the theme of the paper as per the suggestion in the first revision. Now, in the second revision, we have tried our best to improve further and now the manuscript is more focussed as per the theme of the article. Our main theme of the paper involves how cellular circadian metabolism alters in Parkinson’s disease model systems, in particular metabolites and metabolic pathways in PD models. For this, we have added sections, section 2.2: circadian metabolic changes in PD and section 4: challenges of studying circadian changes related to PD in the revised version of manuscript as per the suggestion.  

  1. Additionally, the authors have marked changes in the new version of the manuscript but have not provided a list of the precise changes and the rationale for them.

Response:  

Changes implemented

  • Earlier reviewer had suggested to reduce the section 1, molecular basis of the circadian system, which was actually occupied 9 pages in the initial submitted manuscript. As suggested, we have reduced this section from 9 pages to 3 pages. Specifically, we have highlighted TTFL and metabolism and the rest of sub sections, post-transcriptional modifications that regulate circadian rhythms, post-translational modifications that regulate circadian rhythms and part of TTFL dealing with chromatin modelling and part of circadian control of metabolism were removed as there are specific reviews available for these.
  • Further reviewer asked to improve by stick to the aim of relating PD with circadian metabolism. For this, we have added section 2.2, circadian metabolic changes related to PD, which is actually needed to understand the circadian metabolic perturbations in PD till date. Furthermore, a section 4 dealing with challenges of studying circadian alterations in PD was included as suggested by reviewer, in this we have discussed starting from the selection PD model to delineate altered metabolite profile in PD. These sections were highlighted in red color in the revised manuscript.
  • Further we have added section neural basis of circadian dysfunction as suggested by reviewer 1. That was highlighted in red color.
  1. It would certainly help if the authors identified the challenges of studying circadian changes related to PD and, in particular, the metabolic aspects of such relationship. A critical approach, more than a descriptive one, is needed here. 

Response: Many thanks for the suggestion. We have added a section 2.2, circadian metabolic changes in PD and section 4, titled challenges of studying circadian changes related to PD in the revised version of the manuscript.

While the authors have made some changes, the general legibility of the manuscript could be greatly improved by both a revision of the text and maybe running some kind of readability program, in order to make sure that the reading experience is smooth and (why not) enjoyable.

Thank you for the suggestion. We have taken the help of native English editor for language correction. Furthermore, we have taken the help of Grammarly for improving the English language of the manuscript.
